# Antimicrobial Resistance: KAP of Healthcare Professionals at a Tertiary-Level Hospital in Nepal

**DOI:** 10.3390/ijerph181910062

**Published:** 2021-09-24

**Authors:** Mee-Lang Cheoun, Jongho Heo, Woong-Han Kim

**Affiliations:** 1JW LEE Center for Global Medicine, Seoul National University College of Medicine, 71 Ihwajang-gil, Jongno-gu, Seoul 03087, Korea; meelang95@gmail.com; 2National Assembly Futures Institute, 1 Uisadang-daero, Seoul 07233, Korea; 3Department of Thoracic and Cardiovascular Surgery, Seoul National University College of Medicine, 101 Daehak-ro, Jongno-gu, Seoul 03080, Korea

**Keywords:** antimicrobial resistance, low-middle-income countries, global health, needs assessment, continuing education

## Abstract

Although increasing antimicrobial resistance (AMR) is a substantial threat worldwide, low- and middle-income countries, including Nepal, are especially vulnerable. It is also known that healthcare providers (HCPs) are the major determinants of antimicrobial misuse. A cross-sectional, self-administered survey was conducted among 160 HCPs to assess the knowledge, attitudes, and practices (KAP) of Nepali HCPs regarding AMR and its use. Descriptive statistics and nonparametric tests were performed to evaluate KAP dimensions and investigate subgroup differences. HCPs scored higher on theoretical than practical knowledge. Regarding practical knowledge, men scored higher than women (*p* < 0.01), and physicians scored higher than nurses (*p* < 0.001). Participants aged < 25 years scored lower on practical knowledge than older participants (*p* < 0.001), while those with <3 years work experience scored lower than those with >6 years (*p* < 0.05). Participants from the medical department scored higher on practical knowledge than those from the surgical department (*p* < 0.01). AMR control was more accepted in the medical than in the surgical department (*p* < 0.001). Regarding practices, women and nurses scored higher than men (*p* < 0.001) and physicians (*p* < 0.01), respectively. An educational intervention that is tailored to the sociodemographic and professional characteristics of HCPs is necessary to reduce the gap between theoretical and practical knowledge and improve their attitudes and practices.

## 1. Introduction

Antimicrobial resistance (AMR) is a global concern, affecting both countries with limited resources and developed countries. Since an article was first published in 1996 urging the strengthening of AMR control through antimicrobial stewardship in health systems [1], continuous efforts have been made [1,2]. Most recently, in 2019, AMR was proposed for the first time as a specific indicator of Good Health and Wellbeing, which is Goal 3 of the United Nations’ 2030 Sustainable Development Goals [3]. However, low-middle-income countries (LMICs) with poor infection control and prevention systems are more vulnerable to the increasing threat posed by AMR [4]. The World Bank Report predicted that increasing AMR is estimated to drive an additional 28 million people to extreme poverty by 2050, mainly in LMICs [5].

AMR in Nepal has significantly increased, with an increasing trend in the proportion of multidrug-resistant organisms over the past 20 years [6,7]. In a recent study, more than 50% of *Escherichia coli*, *Klebsiella pneumoniae,* and *Streptococcus pneumoniae* isolates and more than 30% of some *Shigella* sp. and *Vibrio cholerae* isolates were resistant to first-line antibiotics, and many other bacterial pathogens were resistant to most first-line and some second-line antibiotics [8]. The lack of AMR control in the Nepali health systems is attributable to the rarity of bacterial confirmation and susceptibility tests, lack of well-equipped facilities [6], and the low ratio of physicians to patients (0.17 per 1000 population) [9]. These factors may contribute to irrational antibiotic provision, which, in turn, may lead to more self-medication [6] and a high degree of dependency on informal drug dispensers [8], exacerbating the current AMR situation in Nepal.

Inappropriate prescribing patterns of HCPs have been identified as a major determinant of antimicrobial misuse in Nepal [6]; these actions include prescribing broad-spectrum antibiotics at incorrect doses, using antibiotics as a routine course of supportive care, and providing inadequate medication counselling [10]. Given that knowledge, attitudes, and skills are the main factors affecting adequate AMR control [11], HCPs need to increase and update their knowledge and practices on an ongoing basis to keep pace with the constantly changing multisectoral factors surrounding the AMR threat. To the best of our knowledge, KAP studies on AMR targeting HCPs have rarely been conducted in Nepal. Only two KAP studies on AMR were recently conducted among university students [12] and community members [13] in Nepal. Thus, this study aimed to assess the KAP of HCPs toward AMR and the rational use of antibiotics at a tertiary level hospital in Nepal. Additionally, this study aimed to examine the differences in the KAP levels across subgroups categorized by sociodemographic and professional variables.

## 2. Materials and Methods

### 2.1. Study Design and Setting

We conducted a cross-sectional survey among HCPs at a tertiary, nongovernment hospital in south-eastern Kathmandu, Nepal, in January 2020. The university hospital has more than 400 beds, covering the population from the Kavrepalanchok district and other surrounding districts. From August 2020, this institute has been recognized as a COVID-19 care hospital by the government of Nepal, which implies its substantial role in the community.

### 2.2. Study Population

HCPs working at the institute during the study period who were willing to participate in the survey were recruited. HCPs from the Departments of Psychiatry, Ophthalmology, Hair Transplant, Radiology, and Forensics were excluded because they are relatively less involved in the antibiotic process. A minimum sample size of 160 was calculated considering the expected response rate of 50% when a 95% confidence level and a 5% margin of error were applied.

### 2.3. Survey Administration

A self-administered questionnaire was distributed for data collection by co-investigators from both the JW LEE Centre for Global Medicine and the institute during official work hours (08:00–16:00). A total of 200 questionnaires were distributed and only voluntary participants were asked to respond to the questionnaire on site. The study investigators fully explained the purpose of the survey to the participants. Written informed consent was obtained before the survey and 164 (response rate: 82%, 164/200) responses were received. Since four of them missing the demographic information were excluded, the final 160 questionnaires were analysed for this study.

### 2.4. Survey Instrument

The questionnaire was developed based on several related studies [12,14,15,16] and adapted to the Nepali setting (Supplement Document S1). The questionnaire was evaluated by six experts in the field using the content validity index (CVI). For the calculation, we adopted the scale level CVI (SCVI)/Ave, which is defined as the average of the item-level CVI for all items on the scale [17]. A minimum I-CVI of 0.78 and SCVI/Ave of 0.90 indicated excellent content validity [18].

The first domain comprised theoretical knowledge and practical knowledge. This distinction was based on Rolfe’s typology [19], which defines theoretical knowledge as information discovered from books, journals, or lectures, while practical knowledge is defined as information obtained from one’s experience performing relevant tasks in specific situations. With regard to the AMR rate question in the section on practical knowledge, *Klebsiella* spp. was selected not only because it is the second leading bacterial aetiology of healthcare-associated infections (HCAIs) in Nepal [20], but also because it is included in the WHO Global Antimicrobial Resistance Surveillance System (GLASS) list. The true AMR rate was obtained from the GLASS 2017–2018 report [21], and other previous studies were also considered [20].

For the theoretical knowledge domain, HCPs were asked to reply with “True” or “False” to each question. Scores of 1 or 0 were given for the correct or incorrect answers. The practical knowledge domain consisted of two different themes which are ‘Current AMR in the community’ and ‘AMR term familiarity’. For the theme of ’Current AMR in the community’, five different choices including “0–25%”, “25–50%”, “50–75%”, “75–100%”, and “don’t know” were offered. Scores of 1 and 0 were given for the correct and incorrect answers. Lastly, for the theme of ‘AMR term familiarity’, a score of 1 was given to the positive responses: “I’ve heard the term and I can explain what it is”, “I’ve used the term before”, and “I use the term in everyday practice”. A score of 0 to the negative responses: “I’ve never heard of it” and “I’ve heard the term but I’m not sure what it is”.

The second domain included questions on attitudes towards the severity of AMR and the acceptability of AMR interventions that were scored on a 5-point Likert scale. Acceptability was related to education and regulations. The average severity score and the average acceptability score were calculated and ranged from 1 to 5. Additionally, preferences regarding future intervention methods and the sectors perceived as responsible were also ascertained with multiple-choice questions (Appendix A), which were not included in the average acceptability score.

The practice domain incorporated questions asking their management of antibiotics and factors influencing their decisions regarding antibiotic prescriptions and were answered on a 5-point scale with options ranging from “strongly disagree (1)” to “strongly agree (5)”. Based on Bloom’s original cut-off scale, which has been widely used in KAP studies [22,23,24], ≥80% was considered good, ≥60% was considered moderate, and <60% was considered poor when assessing the mean score in each domain.

### 2.5. Statistical Analysis

Descriptive statistics were used to summarize the data. Because the data were not normally distributed by the Shapiro–Wilk test, Mann–Whitney (U) test and Kruskal–Wallis test (*χ*^2^) were used to assess the differences between subgroups stratified by sex, age, work experience, position, and department in each domain (K, A and P). Dunn’s pairwise comparisons were also used for post hoc tests with the Bonferroni adjustment. All analyses were performed using Stata 16.0 (Stata Corp, College Station, TX, USA).

## 3. Results

Table 1 shows the sociodemographic characteristics of the respondents. Overall, nearly twice more women (70.6%) participated as men (29.4%), and half of all the respondents (50.0%) were aged between 25 and 29 years. More than half of the respondents (55.6%) had practiced for less than three years, while 47 (29.4%) had practiced for 3–5 years, and 24 (15.0%) had practiced for more than six years. The respondents included 75 (46.9%) physicians, 76 (47.5%) nurses and 9 (5.6%) pharmacists and pathologists. In total, 151 physicians and nurses were from the medical department, surgical department, anaesthesia department, emergency department, and special care unit. The medical department included internal medicine, paediatrics, and gastrointestinal endoscopy groups, while the surgical department included ear, nose, and throat, obstetrics and gynaecology, orthopaedics, surgery, and post-surgery groups. Nine pathologists and pharmacists were also included in the category of “others” with other physicians and nurses from the anaesthesia department, emergency department, and special care unit.

Overall, the respondents showed a good level of theoretical knowledge (3.51 ± 0.59), while the practical knowledge level was poor (1.35 ± 1.41) (Table 2). Regarding the theoretical knowledge questions, more than 95% of the respondents provided the correct answers to all questions except regarding the antibiotic combinations. In response to the practical knowledge questions concerning the current local AMR of *Klebsiella pneumonia*, only 8 (5%) and 48 (30%) respondents provided the correct answers for ciprofloxacin and meropenem, respectively (Table 3). Regarding ciprofloxacin, almost half (48.8%) of the respondents underestimated, while the other half (46.3%) answered “I do not know”, as shown in Figure 1. The proportion of correct answers was higher for meropenem than for ciprofloxacin, although the proportion of respondents who answered, “I don’t know” (52.5%), was still more than half. Furthermore, most of the participants answered that they had never heard of or were not sure of the practical terms related to AMR (ASP (84.4%) and antibiogram (83.2%)).

The attitudes toward the severity of AMR and acceptability of AMR control were at a moderate level (Table 2). Detailed results regarding the answers to each question are shown in Figure 2. Of all respondents, 83.7% (134/160) considered AMR to be a serious public health issue at the national level, while 61.9% (99/160) considered it to be a serious issue at the facility level. Furthermore, only 30.5% (49/160) of the respondents agreed (combined “Strongly Agree” and “Agree”) that antibiotics were overused in their facility, in contrast to the result that most strongly agreed or agreed with the need to establish education programs regarding rational antibiotic use and antibiotic policies in their facility (79.4% and 81.9%, respectively). 18.8% (30/160) of the respondents believed that limiting the use of antibiotics could impair patient care, and approximately 50.1% of the participants disagreed.

On average, the respondents had a moderate level of accurate practices regarding antibiotics (Table 2). The detailed results regarding the answers to each question are shown in Figure 2. Regarding the related factors, bacterial confirmation and sensitivity reports were considered important prerequisites for antibiotic prescriptions by 71.3% (114/160) of the respondents. Similarly, almost 70% of the respondents agreed that the cost of an antibiotic should be considered before it was prescribed. Nevertheless, 13.3% (10/75) and 17.3% (13/75) of the physicians agreed (combined “Strongly Agree” and “Agree”) that their prescription practices are influenced by patient demands and that they are more influenced by the availability of the antibiotics than by the diseases being treated, respectively.

The Kruskal–Wallis test revealed significant between-sex differences in two domains: practical knowledge (*p* < 0.01) and practices (*p* < 0.001) (Table 4). It also revealed significant differences between age groups (*p* < 0.001) and work experience groups (*p* < 0.05) in practical knowledge. Dunn’s multiple comparisons test showed that respondents younger than 25 years (a) had the lowest level of practical knowledge (a < b, *p* < 0.001; a < c, *p* < 0.001), and the group with less than three years of work experience (a) had significantly less practical knowledge than those with more than six years of work experience (c) (a < c, *p* < 0.05).

Significant differences between physicians and nurses were also found. The physician group (a) had a significantly higher level of practical knowledge than the nurse group (b) (b < a, *p* < 0.001), while the opposite result was observed for practices (a < b, *p* < 0.01).

Finally, between-department differences were found to be significant in three domains: theoretical knowledge (*p* < 0.01), practical knowledge (*p* < 0.001) and the acceptability of AMR control (*p* < 0.001). Respondents in the surgical department (b) had a significantly higher level of theoretical knowledge than those in the other departments (c < b, *p* < 0.001), while the opposite result was observed for practical knowledge (b < c, *p* < 0.05). Regarding the acceptability of the AMR control, a significant difference was observed not only between the medical department (a) and other departments (c) (c < a, *p* < 0.01) but also between the medical department (a) and surgical department (b) (b < a, *p* < 0.001).

Regarding the method of delivery of educational interventions preferred by HCPs (Appendix A), their first choice was including AMR-related content into the official undergraduate curricula of medical and nursing schools (50.9%). An internal training program offered by the institute was preferred (36.5%) over continuing medical education (CME) programs offered by national institutions (27.0%) or international organizations (22.6%).

## 4. Discussion

Using descriptive statistics and nonparametric analysis, we examined the current status of KAP and differences across the subgroups of HCPs. On average, the HCPs showed a high level of theoretical knowledge (3.51/4) and a low level of practical knowledge (1.35/6). The level of agreement with the concept that AMR is a severe problem (mean: 3.62; range: 1–5) and the level of agreement with AMR control measures (mean: 3.94; range: 1–5) was both moderate. Regarding practice, HCPs had a moderate level of correct practices regarding antibiotic management and prescriptions (mean: 3.91; range 1–5).

Additionally, we found significant differences in the KAP dimensions across subgroups. The oldest age group had significantly greater practical knowledge than the youngest age group (mean rank: 93.43 vs. 49.15; *p* < 0.001). HCPs with more than six years of work experience had significantly more practical knowledge than those with three years of work experience (mean rank: 98.57 vs. 71.73; *p* < 0.05). The physician group had significantly more practical knowledge than the nurse group (mean rank: 93.23 vs. 63.86; *p* < 0.001), while the opposite result was found for practices (mean rank: 69.81 vs. 91.63; *p* < 0.01). The medical department respondents had higher levels of practical knowledge (mean rank: 101.86 vs. 61.31; *p* < 0.001) and acceptance of the AMR control measures (mean rank: 92.44 vs. 60.78; *p* < 0.001) than the surgical department respondents. Across all five categories, no significant differences were observed between the subgroups in attitudes toward the severity of AMR.

Although on average, HCPs had a good level of theoretical knowledge regarding the definition of AMR and clinical indications for antibiotic use, most did not know the current AMR rate in their community. A previous KAP study on AMR in Peru showed that only 20% of the participants correctly estimated the level of resistance of *K. pneumonia* to cephalosporins [14], a finding that is similar to that in our study. Interestingly, the proportion of correct answers for ciprofloxacin was lower than that for meropenem in our study. This finding implies that the AMR risk of the more commonly used antibiotic, against which the second leading pathogen responsible for HCAIs in Nepal has a higher rate of resistance, is now being underestimated by HCPs. The ciprofloxacin resistance rates of *Klebsiella* spp [20,22]. and *K. pneumonia* [21] are both higher than the meropenem resistance rates in Nepal, while ciprofloxacin is one of the top five antibiotics most commonly dispensed by private pharmacies in Nepal [25].

The low level of familiarity with the AMR-related practical terms among HCPs should also be noted. ASP is one of the most basic and well-known methods of decreasing antibiotic resistance [26]. However, in Nepal, no formal efforts regarding stewardship had been undertaken before 2017 [27].

In the attitude dimension, AMR was generally perceived among HCPs as a serious public health issue in Nepal; meanwhile, AMR was much less well recognized as a serious issue within their own facility. Notably, approximately one-fifth of the HCPs considered limiting the use of antibiotics to be an obstacle to patient care. Regarding the inconsistent perception of the severity of the problem between the national and facility levels, previous KAP studies on AMR have reported similar findings [14,28]. These results may imply that respondents perceive AMR to be a theoretical and text-based problem rather than a real and tangible problem, possibly creating a barrier to future changes in behaviour [28]. Krockow et al. [29] further explained that the lack of imminence and abstract nature of the threat posed by AMR might lead staff to believe that neither they nor their patients are at risk. Considering this lower perceived severity at their own facility level, each institute might consider some collaborative work with external institutes, at least in its early stages, to elicit their voluntary change based on a reasonably objective understanding of their current status.

The level of negative attitudes toward the restrictive measures was similar (19.1%) in a previous KAP study in Iran [16]. A previous qualitative study conducted in Sweden showed that a physician’s uncertainty regarding managing infectious diseases is a barrier to restrictions on antibiotic use [30].

In the practice dimension, this study found two factors influencing HCPs’ decisions regarding antibiotic management and prescriptions. Approximately 70% of the participants considered bacterial confirmation tests, sensitivity reports, and the cost of an antibiotic as important prerequisites to writing prescriptions. Bacterial identification and susceptibility tests can reduce the empiric prescribing of broad-spectrum antibiotics [31], and they are highly recommended to maintain the susceptibility of current antimicrobials for as long as possible [2]. However, in resource-limited settings, establishing laboratory facilities for culturing and sensitivity testing may not always be possible because of the associated expense [32]. Even when well-equipped facilities are established, practical availability is another problem. Many barriers may exist preventing them from being utilised, such as financial constraints for the patients, the lack of routine blood sample collection for culturing, the lack of experienced personnel, and a long-time lag between testing and results [33]. Given that this study was conducted at a tertiary level setting in a semi-urban area of Nepal, the empirical use of antibiotics may be even more common in peripheral health facilities.

The cost of drugs has been reported to be an affecting factor to the prescriptions written by physicians [16,34,35], and our study also showed a similar result. Minimizing the financial burden on patients in LMICs has been reported to be essential to ensuring that patients are using qualified antibiotics rather than relying on unregulated drug providers [36]. Especially, in Nepal, the out-of-pocket (OOP) expenditure for healthcare was shown to account for 55.4% of the total current health expenditure in 2015, much higher than the proportions of other Asian countries and the level of 15–20% recommended by WHO [37]. An analysis of 47 LMICs showed that a 10% increase in OOP health expenditures is associated with a 3.2% increase in AMR [38]. Consequently, to prevent people relying on unregulated drug dispensaries and borrowing medicines from their neighbours which are often unknown chemicals and substandard doses [36], more emphasis on the price of antibiotics entwined with poverty [36] and other economic factors such as local coverage or co-payment strategies should be considered for their prescription decision-making. [35] If physicians were able to access therapeutically equivalent but less-expensive drugs, a large burden of patients could be saved and they would not delay timely treatment [35] which is very important for controlling AMR.

In our study, the youngest age group and the group with the least work experience had significantly lower levels of practical knowledge than their counterparts, a finding that is similar to the result of a study in Ghana [15]. The older groups’ higher professional standing and longer experience may explain their better practical knowledge. However, no significant difference was observed between those subgroups in terms of theoretical knowledge. This inconsistent knowledge deficit can be partly explained by the fact that AMR has become an important issue in the last 10–15 years, while older HCPs who had finished training before that period have limited theoretical knowledge on this topic [39]. In this regard, apart from reinforcing AMR content in the official undergraduate curricula of medical and nursing schools (Appendix A), keeping working HCPs equipped with the latest AMR knowledge and information also seems necessary. When a continuing AMR education program is organized for HCPs in the field, different focuses on theoretical and practical knowledge should be emphasized according to HCPs’ ages and years of work experience.

The practical knowledge level of the surgical department in our study was the lowest among the subgroups. In surgery, the prescription of antibiotics is usually made based on biological markers such as the C-reactive protein level and white cell count [40], rather than DDD, DOT and antibiograms. In other words, surgical department physicians focus more on the prophylactic use of antibiotics than on avoiding the prolonged use of antibiotics to prevent AMR [40,41,42,43].

Opposite results in the comparison of the physician and nurse groups were observed regarding practical knowledge and practices, and a similar result was reported in India [44], although that study did not distinguish between theoretical and practical knowledge. Considering that our study was conducted using a self-administered questionnaire, these findings may imply that physicians who have antibiotic prescription authority have higher standards when evaluating their own practices.

Additionally, the medical department respondents had a significantly higher level of acceptance of AMR control measures than the surgical department and other departments. Concerning the surgical department HCPs who have prophylactic prescribing patterns and use broad-spectrum antibiotics to avert the risk of future infections [26], any external measure to control AMR would be less acceptable.

Regarding the educational method, the preference for internal educator and face-to-face instruction was observed (Appendix A). However, though external (27.0%) and international educators (22.6%) were not highly preferred by general HCPs from our study, it seems inevitable to cooperate with them considering that global health partnership (GHP) has been considered as one of the best approaches for strengthening healthcare capacity [45]. Instead, adopting a “cascading approach” by GHP for fostering educators to educate HCPs internally may be effective in terms of swiftness and self-sustainability [45].

This study has limitations. First, the provision of socially desirable answers by the participants may have contributed to bias; this is a common issue in studies using self-reported data. Second, this study was performed at a tertiary-level teaching hospital covering more than 5 different districts, and the results cannot necessarily be generalized to all health care settings.

## 5. Conclusions

The present study showed a knowledge deficit in the use of combined antibiotics, practical AMR terms, and current AMR rates among HCPs in Nepal. Significant differences in KAP dimensions across subgroups were also found. Our study suggests that educational programs targeting the working HCPs for their continuous learning need to focus on closing the gap between theoretical and practical knowledge and reinforcing better attitudes and practices regarding AMR and rational antibiotic use. Especially for the practical knowledge, which was the weakest dimension from our study; more emphasis should be put on surgical departments and younger professionals. Curricula that are carefully tailored to the differences in the KAP dimensions according to sociodemographic and professional characteristics may increase the acceptance and effectiveness of the programs.

## Figures and Tables

**Figure 1 ijerph-18-10062-f001:**
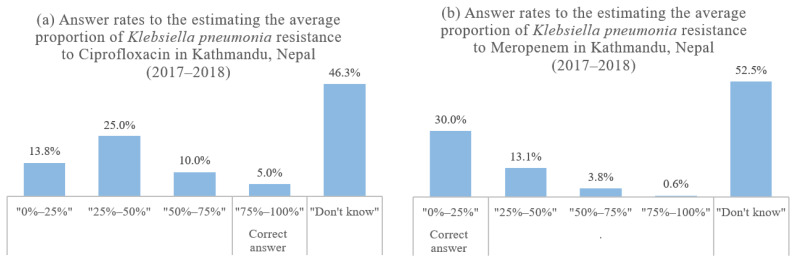
Answer rates to the questions of current AMR rates in the community: *Klebsiella pneumonia* to (**a**) ciprofloxacin and (**b**) meropenem in Kathmandu, Nepal (*n* = 160).

**Figure 2 ijerph-18-10062-f002:**
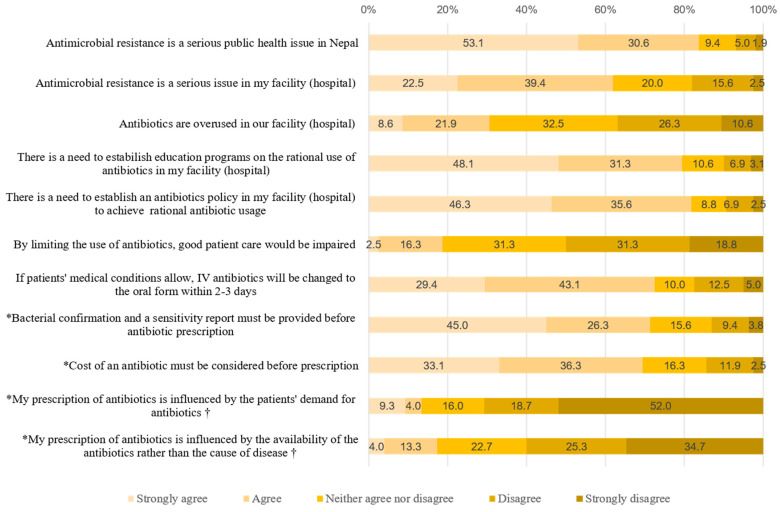
Results of the attitudes and practices test (*n* = 160). * Questions regarding the factors that influence antibiotic prescription. † asked only of *physicians* who have prescribing authority.

**Table 1 ijerph-18-10062-t001:** Demographic characteristics of the participants (*n* = 160).

Category	Group	*n* (%)
Sex	Male	47 (29.4)
	Female	113 (70.6)
Age (years)	<25 years	44 (27.5)
	25 years ≤ … < 30 years	80 (50.0)
	≥30 years	36 (22.5)
Experience (years)	<3 years	89 (55.6)
	3 years ≤ … < 6 years	47 (29.4)
	≥6 years	24 (15.0)
Position	Physician	75 (46.9)
	Nurse	76 (47.5)
	Others	9 (5.6)
Department	Medicine	40 (25.0)
	Internal Medicine	29 (18.1)
	Pediatrics	6 (3.8)
	Gastrointestinal Endoscopy	5 (3.1)
	Surgical	65 (40.6)
	Ear, Nose, and Throat	9 (5.6)
	Obstetrics and Gynecology	21 (13.1)
	Orthopedics	18 (11.3)
	Operating Theater	2 (1.3)
	Post-Surgery	15 (9.4)
	Others	55 (34.4)
	Anesthesia	11 (6.9)
	Emergency	8 (5.0)
	CCU, ICU, and HDU	27 (16.9)
	Pathology	4 (2.5)
	Pharmacy	5 (3.1)

CCU: cardiac care unit, ICU: intensive care unit, HDU: high-dependency unit.

**Table 2 ijerph-18-10062-t002:** Mean and median scores for knowledge, attitudes, and practices.

Domain		Mean Score ± SD	Median Score (IQR)
Knowledge	Theoretical	3.51 ± 0.59	4 (3–4)
	Practical	1.35 ± 1.41	1 (0–2)
Attitude	Severity	3.62 ± 0.82	4 (3–4.33)
	Acceptability	3.94 ± 0.70	4 (3.67–4.33)
Practice		3.91 ± 0.73	4 (3.40–4.33)

**Table 3 ijerph-18-10062-t003:** Results of the knowledge test (*n* = 160).

Domain/Theme		Question				*n* (%) of Correct Answers	
Theoretical Knowledge		Antimicrobial resistance indicates that bacteria develop the ability to defeat the antibiotics designed to kill them.				154 (96.3)	
		Patients with common cold symptoms need antibiotic treatment.				153 (95.6)	
		Antibiotics cure viral infections.				155 (96.9)	
		Combinations of antibiotics can help prevent antimicrobial resistance.				100 (62.5)	
Practical Knowledge	Current AMR in community	Average proportion of *Klebsiella pneumonia* resistant to ciprofloxacin in Kathmandu, Nepal (2017–2018).				8 (5.0)	
		Average proportion of *Klebsiella pneumonia* resistant to meropenem in Kathmandu, Nepal (2017–2018).				48 (30.0)	
	Term familiarity	**Question**	** *n* ** **(%) of responses**				
			Never heard	Heard but not sure	Heard and can explain	Have used Before	Use every-day
		Antibiotics Stewardship Program	100 (62.5)	35 (21.9)	12 (7.5)	8 (5.0)	5 (3.1)
		Defined Daily Dose	64 (40.0)	53 (33.1)	18 (11.3)	12 (7.5)	13 (8.1)
		Days of Therapy	43 (26.9)	52 (32.5)	31 (19.4)	17 (10.6)	17 (10.6)
		Antibiogram	94 (58.8)	39 (24.4)	13 (8.1)	6 (3.8)	8 (5.0)

**Table 4 ijerph-18-10062-t004:** Group differences in knowledge, attitudes and practices (*n* = 160).

	Knowledge	Attitudes	Practices
Theoretical Knowledge	Practical Knowledge	Seriousness	Acceptability
Category	Group	Mean Rank	U *or χ*^2^	Post hoc^†††^	Mean Rank	U *or χ*^2^	Post hoc^†††^	Mean Rank	U *or χ*^2^	Post hoc ^†††^	Mean Rank	U *or χ*^2^	Post hoc^†††^	Mean Rank	U *or χ*^2^	Post hoc ^†††^
Sex ^†^	Male	86.51	1291.5		96.61	817 **		86.15	1308.5		77.21	1728.5		61.56	2464 ***	
Female	78	73.80	78.15	81.87	88.38
Age ^††^	<25 years ^a^	81.02	0.373		49.15	26.563 ***	a < c ***a < b ***	65.82	5.094		74.85	0.409		80.45	1.097	
25 years≤ <30 years ^b^	78.48	87.48	81.23	79.42	74.99
30 years≤ ^c^	75.51	93.43	87.56	80.8	84.07
Experience ^††^	<3 years ^a^	84.30	4.483		71.73	7.126 *	a < c *	72.20	4.318		77.31	0.167		79.03	1.636	
3 years≤ <6 years ^b^	73.26	81.06	83.54	8061	73.07
6 years≤ ^c^	67.52	98.57	91.76	78.65	87.59
Position ^††^	Physician ^a^	83.51	2.347		93.23	16.926 ***	b < a ***	86	3.128		80.69	0.04		69.81	8.561 *	a < b **
Nurse ^b^	73.68	63.86	77.45	80	91.63
Others ^c^	75.5	73.94	60.44	83.11	75.61
Department ^††^	Medical ^a^	77.50	19.154 **	c < b ***	101.86	21.344 ***	b < a ***c < a *b < c *	83.99	5.316		92.44	16.430 ***	b < a ***c < a **	88.54	3.739	
Surgical ^b^	93.75	61.31	68.37	60.78	79.05
Others ^c^	61.69	80.93	86.06	88.52	70.44

## Data Availability

The datasets for this study are available from the corresponding authors on reasonable request.

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
