# Peer review of "Antimicrobial Resistance: KAP of Healthcare Professionals at a Tertiary-Level Hospital in Nepal"

_ijerph, 2021, doi:10.3390/ijerph181910062_

Round 1

Reviewer 1 Report

The Materials and Methods section needs to be edited to make a few aspects clearer. The following issues are raised as they appear in the text rather than by importance:  

1) L79: Please add 'response' between expected and rate.

 2) L83: Is the Institute the university teaching hospital?

3) L87: Mention made that the questionnaire was developed based on several related studies, yet reference is only given to one study.

4) L103: The theoretical knowledge questions were stated to have a Yes/No answer. Table 2 suggests a (0-4) scale? And the Lines proceeding this (L104-112) are difficult to follow. 

5) L125: Overall mean score is not the correct term. Do you mean cumulative response rate or similar? 

Results

6) Please provide response rate. How representative were the respondents of HCPs working at the hospital? It seems a very young cohort. 

7) L149: Suggests the Yes/No response stipulated in the methods was incorrect and the (0-4) in Table 2 was correct?

8) Table 4 - Not clear how the Kruskal-Wallis test is being applied. For example, if the mean score for practical knowledge was 1.35, then how are the mean ranks so high? And how can a mean rank be 101.86 for the medical department?

General

9) Note the supplementary Figure 1 was unavailable to reviewers. 

10) Please add the questionnaire as a supplementary appendix. This would make the methods and write-up of the results easier to follow. 

Reviewer 2 Report

I congratulate the authors for the topic approached, the way of exploring and explaining the results.

I have no observations, but

Methods:  Which studies are used for the development of questionnaire?

Results:

-figure 1

-table 2 is difficult to understand; the answer is yes/no or a scale 0-4?

Discussion: can be improved by

- given the results of the study, what measures can be taken?

- in case of improving the medical knowledge, who will be the target group?

- how the costs of antibiotic therapy can be incorporated into therapeutic decision making

-if is possible, reconfigure figure 1 

-it is possible other references to be added? 
